# Weight Normalization: A Simple Reparameterization to Accelerate Training of Deep Neural Networks

**Tim Salimans**
OpenAI
tim@openai.com

**Diederik P. Kingma**
OpenAI
dpkingma@openai.com

## Abstract

We present *weight normalization*: a reparameterization of the weight vectors in a neural network that decouples the length of those weight vectors from their direction. By reparameterizing the weights in this way we improve the conditioning of the optimization problem and we speed up convergence of stochastic gradient descent. Our reparameterization is inspired by batch normalization but does not introduce any dependencies between the examples in a minibatch. This means that our method can also be applied successfully to recurrent models such as LSTMs and to noise-sensitive applications such as deep reinforcement learning or generative models, for which batch normalization is less well suited. Although our method is much simpler, it still provides much of the speed-up of full batch normalization. In addition, the computational overhead of our method is lower, permitting more optimization steps to be taken in the same amount of time. We demonstrate the usefulness of our method on applications in supervised image recognition, generative modelling, and deep reinforcement learning.

## 1   Introduction

Recent successes in deep learning have shown that neural networks trained by first-order gradient based optimization are capable of achieving amazing results in diverse domains like computer vision, speech recognition, and language modelling [7]. However, it is also well known that the practical success of first-order gradient based optimization is highly dependent on the curvature of the objective that is optimized. If the condition number of the Hessian matrix of the objective at the optimum is low, the problem is said to exhibit *pathological curvature*, and first-order gradient descent will have trouble making progress [22, 32]. The amount of curvature, and thus the success of our optimization, is not invariant to reparameterization [1]: there may be multiple equivalent ways of parameterizing the same model, some of which are much easier to optimize than others. Finding good ways of parameterizing neural networks is thus an important problem in deep learning.

While the architectures of neural networks differ widely across applications, they are typically mostly composed of conceptually simple computational building blocks sometimes called neurons: each such neuron computes a *weighted sum* over its inputs and adds a *bias* term, followed by the application of an elementwise nonlinear transformation. Improving the general optimizability of deep networks is a challenging task [6], but since many neural architectures share these basic building blocks, improving these building blocks improves the performance of a very wide range of model architectures and could thus be very useful.

Several authors have recently developed methods to improve the conditioning of the cost gradient for general neural network architectures. One approach is to explicitly left multiply the cost gradient with an approximate inverse of the Fisher information matrix, thereby obtaining an approximately whitened *natural gradient*. Such an approximate inverse can for example be obtained by using a Kronecker factored approximation to the Fisher matrix and inverting it (KFAC, [23]), by using an

approximate Cholesky factorization of the inverse Fisher matrix (FANG, [10]), or by whitening the input of each layer in the neural network (PRONG, [5]).

Alternatively, we can use standard first order gradient descent without preconditioning, but change the parameterization of our model to give gradients that are more like the whitened natural gradients of these methods. For example, Raiko et al. [27] propose to transform the outputs of each neuron to have zero output and zero slope on average. They show that this transformation approximately diagonalizes the Fisher information matrix, thereby whitening the gradient, and that this leads to improved optimization performance. Another approach in this direction is *batch normalization* [14], a method where the output of each neuron (before application of the nonlinearity) is normalized by the mean and standard deviation of the outputs calculated over the examples in the minibatch. This reduces covariate shift of the neuron outputs and the authors suggest it also brings the Fisher matrix closer to the identity matrix.

Following this second approach to approximate natural gradient optimization, we propose a simple but general method, called *weight normalization*, for improving the optimizability of the weights of neural network models. The method is inspired by batch normalization, but it is a deterministic method that does not share batch normalization's property of adding noise to the gradients. In addition, the overhead imposed by our method is lower: no additional memory is required and the additional computation is negligible. The method show encouraging results on a wide range of deep learning applications.

## 2 Weight Normalization

We consider standard artificial neural networks where the computation of each neuron consists in taking a weighted sum of input features, followed by an elementwise nonlinearity:

$$y = \phi(\mathbf{w} \cdot \mathbf{x} + b), \tag{1}$$

where $\mathbf{w}$ is a $k$-dimensional weight vector, $b$ is a scalar bias term, $\mathbf{x}$ is a $k$-dimensional vector of input features, $\phi(.)$ denotes an elementwise nonlinearity such as the rectifier $\max(.,0)$, and $y$ denotes the scalar output of the neuron.

After associating a loss function to one or more neuron outputs, such a neural network is commonly trained by stochastic gradient descent in the parameters $\mathbf{w}, b$ of each neuron. In an effort to speed up the convergence of this optimization procedure, we propose to reparameterize each weight vector $\mathbf{w}$ in terms of a parameter vector $\mathbf{v}$ and a scalar parameter $g$ and to perform stochastic gradient descent with respect to those parameters instead. We do so by expressing the weight vectors in terms of the new parameters using

$$\mathbf{w} = \frac{g}{||\mathbf{v}||}\mathbf{v} \tag{2}$$

where $\mathbf{v}$ is a $k$-dimensional vector, $g$ is a scalar, and $||\mathbf{v}||$ denotes the Euclidean norm of $\mathbf{v}$. This reparameterization has the effect of fixing the Euclidean norm of the weight vector $\mathbf{w}$: we now have $||\mathbf{w}|| = g$, independent of the parameters $\mathbf{v}$. We therefore call this reparameterizaton *weight normalization*.

The idea of normalizing the weight vector has been proposed before (e.g. [31, 33]) but earlier work typically still performed optimization in the $\mathbf{w}$-parameterization, only applying the normalization after each step of stochastic gradient descent. This is fundamentally different from our approach: we propose to explicitly reparameterize the model and to perform stochastic gradient descent in the new parameters $\mathbf{v}, g$ directly. Doing so improves the conditioning of the gradient and leads to improved convergence of the optimization procedure: By decoupling the norm of the weight vector ($g$) from the direction of the weight vector ($\mathbf{v}/||\mathbf{v}||$), we speed up convergence of our stochastic gradient descent optimization, as we show experimentally in section 5.

Instead of working with $g$ directly, we may also use an exponential parameterization for the scale, i.e. $g = e^s$, where $s$ is a log-scale parameter to learn by stochastic gradient descent. Parameterizing the $g$ parameter in the log-scale is more intuitive and more easily allows $g$ to span a wide range of different magnitudes. Empirically, however, we did not find this to be an advantage. In our experiments, the eventual test-set performance was not significantly better or worse than the results with directly learning $g$ in its original parameterization, and optimization was slightly slower.

## 2.1 Gradients

Training a neural network in the new parameterization is done using standard stochastic gradient descent methods. Here we differentiate through (2) to obtain the gradient of a loss function $L$ with respect to the new parameters $\mathbf{v}, g$. Doing so gives

$$\nabla_g L = \frac{\nabla_{\mathbf{w}} L \cdot \mathbf{v}}{||\mathbf{v}||}, \qquad \nabla_{\mathbf{v}} L = \frac{g}{||\mathbf{v}||} \nabla_{\mathbf{w}} L - \frac{g \nabla_g L}{||\mathbf{v}||^2} \mathbf{v}, \tag{3}$$

where $\nabla_{\mathbf{w}} L$ is the gradient with respect to the weights $\mathbf{w}$ as used normally.

Backpropagation using weight normalization thus only requires a minor modification to the usual backpropagation equations, and is easily implemented using standard neural network software, either by directly specifying the network in terms of the $\mathbf{v}, g$ parameters and relying on auto-differentiation, or by applying (3) in a post-processing step. We provide reference implementations using both approaches for Theano, Tensorflow and Keras at `https://github.com/openai/weightnorm`. Unlike with batch normalization, the expressions above are independent of the minibatch size and thus cause only minimal computational overhead.

An alternative way to write the gradient is

$$\nabla_{\mathbf{v}} L = \frac{g}{||\mathbf{v}||} M_{\mathbf{w}} \nabla_{\mathbf{w}} L, \quad \text{with} \quad M_{\mathbf{w}} = I - \frac{\mathbf{w}\mathbf{w}'}{||\mathbf{w}||^2}, \tag{4}$$

where $M_{\mathbf{w}}$ is a projection matrix that projects onto the complement of the $\mathbf{w}$ vector. This shows that weight normalization accomplishes two things: it *scales* the weight gradient by $g/||\mathbf{v}||$, and it *projects* the gradient away from the current weight vector. Both effects help to bring the covariance matrix of the gradient closer to identity and benefit optimization, as we explain below.

Due to projecting away from $\mathbf{w}$, the norm of $\mathbf{v}$ grows monotonically with the number of weight updates when learning a neural network with weight normalization using standard gradient descent without momentum: Let $\mathbf{v}' = \mathbf{v} + \Delta\mathbf{v}$ denote our parameter update, with $\Delta\mathbf{v} \propto \nabla_{\mathbf{v}} L$ (steepest ascent/descent), then $\Delta\mathbf{v}$ is necessarily orthogonal to the current weight vector $\mathbf{w}$ since we project away from it when calculating $\nabla_{\mathbf{v}} L$ (equation 4). Since $\mathbf{v}$ is proportional to $\mathbf{w}$, the update is thus also orthogonal to $\mathbf{v}$ and increases its norm by the Pythagorean theorem. Specifically, if $||\Delta\mathbf{v}||/||\mathbf{v}|| = c$ the new weight vector will have norm $||\mathbf{v}'|| = \sqrt{||\mathbf{v}||^2 + c^2||\mathbf{v}||^2} = \sqrt{1 + c^2}||\mathbf{v}|| \geq ||\mathbf{v}||$. The rate of increase will depend on the the variance of the weight gradient. If our gradients are noisy, $c$ will be high and the norm of $\mathbf{v}$ will quickly increase, which in turn will decrease the scaling factor $g/||\mathbf{v}||$. If the norm of the gradients is small, we get $\sqrt{1 + c^2} \approx 1$, and the norm of $\mathbf{v}$ will stop increasing. Using this mechanism, the scaled gradient self-stabilizes its norm. This property does not strictly hold for optimizers that use separate learning rates for individual parameters, like Adam [15] which we use in experiments, or when using momentum. However, qualitatively we still find the same effect to hold.

Empirically, we find that the ability to grow the norm $||\mathbf{v}||$ makes optimization of neural networks with weight normalization very robust to the value of the learning rate: If the learning rate is too large, the norm of the unnormalized weights grows quickly until an appropriate effective learning rate is reached. Once the norm of the weights has grown large with respect to the norm of the updates, the effective learning rate stabilizes. Neural networks with weight normalization therefore work well with a much wider range of learning rates than when using the normal parameterization. It has been observed that neural networks with batch normalization also have this property [14], which can also be explained by this analysis.

By projecting the gradient away from the weight vector $\mathbf{w}$, we also eliminate the noise in that direction. If the covariance matrix of the gradient with respect to $\mathbf{w}$ is given by $\mathbf{C}$, the covariance matrix of the gradient in $\mathbf{v}$ is given by $\mathbf{D} = (g^2/||\mathbf{v}||^2) M_{\mathbf{w}} \mathbf{C} M_{\mathbf{w}}$. Empirically, we find that $\mathbf{w}$ is often (close to) a dominant eigenvector of the covariance matrix $\mathbf{C}$: removing that eigenvector then gives a new covariance matrix $\mathbf{D}$ that is closer to the identity matrix, which may further speed up learning.

## 2.2 Relation to batch normalization

An important source of inspiration for this reparameterization is *batch normalization* [14], which normalizes the statistics of the pre-activation $t$ for each minibatch as

$$t' = \frac{t - \mu[t]}{\sigma[t]},$$

with $\mu[t], \sigma[t]$ the mean and standard deviation of the pre-activations $t = \mathbf{v} \cdot \mathbf{x}$. For the special case where our network only has a single layer, and the input features $\mathbf{x}$ for that layer are whitened (independently distributed with zero mean and unit variance), these statistics are given by $\mu[t] = 0$ and $\sigma[t] = ||\mathbf{v}||$. In that case, normalizing the pre-activations using batch normalization is equivalent to normalizing the weights using weight normalization.

Convolutional neural networks usually have much fewer weights than pre-activations, so normalizing the weights is often much cheaper computationally. In addition, the norm of $\mathbf{v}$ is non-stochastic, while the minibatch mean $\mu[t]$ and variance $\sigma^2[t]$ can in general have high variance for small minibatch size. Weight normalization can thus be viewed as a cheaper and less noisy approximation to batch normalization. Although exact equivalence does not usually hold for deeper architectures, we still find that our weight normalization method provides much of the speed-up of full batch normalization. In addition, its deterministic nature and independence on the minibatch input also means that our method can be applied more easily to models like RNNs and LSTMs, as well as noise-sensitive applications like reinforcement learning.

## 3 Data-Dependent Initialization of Parameters

Besides a reparameterization effect, batch normalization also has the benefit of fixing the scale of the features generated by each layer of the neural network. This makes the optimization robust against parameter initializations for which these scales vary across layers. Since weight normalization lacks this property, we find it is important to properly initialize our parameters. We propose to sample the elements of $\mathbf{v}$ from a simple distribution with a fixed scale, which is in our experiments a normal distribution with mean zero and standard deviation 0.05. Before starting training, we then initialize the $b$ and $g$ parameters to fix the minibatch statistics of all pre-activations in our network, just like in batch normalization, but only for a single minibatch of data and only during initialization. This can be done efficiently by performing an initial feedforward pass through our network for a single minibatch of data $\mathbf{X}$, using the following computation at each neuron:

$$t = \frac{\mathbf{v} \cdot \mathbf{x}}{||\mathbf{v}||}, \quad \text{and} \quad y = \phi\left(\frac{t - \mu[t]}{\sigma[t]}\right), \tag{5}$$

where $\mu[t]$ and $\sigma[t]$ are the mean and standard deviation of the pre-activation $t$ over the examples in the minibatch. We can then initialize the neuron's bias $b$ and scale $g$ as

$$g \leftarrow \frac{1}{\sigma[t]}, \qquad b \leftarrow \frac{-\mu[t]}{\sigma[t]}, \tag{6}$$

so that $y = \phi(\mathbf{w} \cdot \mathbf{x} + b)$. Like batch normalization, this method ensures that all features initially have zero mean and unit variance before application of the nonlinearity. With our method this only holds for the minibatch we use for initialization, and subsequent minibatches may have slightly different statistics, but experimentally we find this initialization method to work well. The method can also be applied to networks without weight normalization, simply by doing stochastic gradient optimization on the parameters $\mathbf{w}$ directly, after initialization in terms of $\mathbf{v}$ and $g$: this is what we compare to in section 5. Independently from our work, this type of initialization was recently proposed by different authors [24, 18] who found such data-based initialization to work well for use with the standard parameterization in terms of $\mathbf{w}$.

The downside of this initialization method is that it can only be applied in similar cases as where batch normalization is applicable. For models with recursion, such as RNNs and LSTMs, we will have to resort to standard initialization methods.

## 4   Mean-only Batch Normalization

Weight normalization, as introduced in section 2, makes the scale of neuron activations approximately independent of the parameters $\mathbf{v}$. Unlike with batch normalization, however, the means of the neuron activations still depend on $\mathbf{v}$. We therefore also explore the idea of combining weight normalization with a special version of batch normalization, which we call *mean-only batch normalization*: With this normalization method, we subtract out the minibatch means like with full batch normalization, but we do not divide by the minibatch standard deviations. That is, we compute neuron activations using

$$t = \mathbf{w} \cdot \mathbf{x}, \qquad \tilde{t} = t - \mu[t] + b, \qquad y = \phi(\tilde{t}) \tag{7}$$

where $\mathbf{w}$ is the weight vector, parameterized using weight normalization, and $\mu[t]$ is the minibatch mean of the pre-activation $t$. During training, we keep a running average of the minibatch mean which we substitute in for $\mu[t]$ at test time.

The gradient of the loss with respect to the pre-activation $t$ is calculated as

$$\nabla_t L = \nabla_{\tilde{t}} L - \mu[\nabla_{\tilde{t}} L], \tag{8}$$

where $\mu[.]$ denotes once again the operation of taking the minibatch mean. Mean-only batch normalization thus has the effect of centering the gradients that are backpropagated. This is a comparatively cheap operation, and the computational overhead of mean-only batch normalization is thus lower than for full batch normalization. In addition, this method causes less noise during training, and the noise that is caused is more gentle as the law of large numbers ensures that $\mu[t]$ and $\mu[\nabla_{\tilde{t}}]$ are approximately normally distributed. Thus, the added noise has much lighter tails than the highly kurtotic noise caused by the minibatch estimate of the variance used in full batch normalization. As we show in section 5.1, this leads to improved accuracy at test time.

## 5   Experiments

We experimentally validate the usefulness of our method using four different models for varied applications in supervised image recognition, generative modelling, and deep reinforcement learning.

### 5.1   Supervised Classification: CIFAR-10

To test our reparameterization method for the application of supervised classification, we consider the CIFAR-10 data set of natural images [19]. The model we are using is based on the ConvPool-CNN-C architecture of [30], with some small modifications: we replace the first dropout layer by a layer that adds Gaussian noise, we expand the last hidden layer from 10 units to 192 units, and we use $2 \times 2$ max-pooling, rather than $3 \times 3$. The only hyperparameter that we actively optimized (the standard deviation of the Gaussian noise) was chosen to maximize the performance of the network on a holdout set of 10000 examples, using the standard parameterization (no weight normalization or batch normalization). A full description of the resulting architecture is given in table A in the supplementary material.

We train our network for CIFAR-10 using Adam [15] for 200 epochs, with a fixed learning rate and momentum of 0.9 for the first 100 epochs. For the last 100 epochs we set the momentum to 0.5 and linearly decay the learning rate to zero. We use a minibatch size of 100. We evaluate 5 different parameterizations of the network: 1) the standard parameterization, 2) using batch normalization, 3) using weight normalization, 4) using weight normalization combined with *mean-only* batch normalization, 5) using mean-only batch normalization with the normal parameterization. The network parameters are initialized using the scheme of section 3 such that all four cases have identical parameters starting out. For each case we pick the optimal learning rate in $\{0.0003, 0.001, 0.003, 0.01\}$. The resulting error curves during training can be found in figure 1: both weight normalization and batch normalization provide a significant speed-up over the standard parameterization. Batch normalization makes slightly more progress per epoch than weight normalization early on, although this is partly offset by the higher computational cost: with our implementation, training with batch normalization was about 16% slower compared to the standard parameterization. In contrast, weight normalization was not noticeably slower. During the later stage of training, weight normalization and batch normalization seem to optimize at about the same speed, with the normal parameterization (with or without mean-only batch normalization) still lagging behind. After optimizing the network for 200

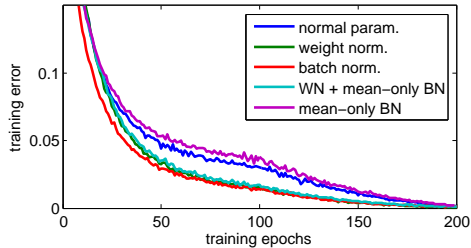

Figure 1: Training error for CIFAR-10 using different parameterizations. For *weight normalization*, *batch normalization*, and *mean-only batch normalization* we show results using Adam with a learning rate of 0.003. For the normal parameterization we instead use 0.0003 which works best in this case. For the last 100 epochs the learning rate is linearly decayed to zero.

| Model | Test Error |
|---|---|
| Maxout [8] | 11.68% |
| Network in Network [21] | 10.41% |
| Deeply Supervised [20] | 9.6% |
| ConvPool-CNN-C [30] | 9.31% |
| ALL-CNN-C [30] | 9.08% |
| our CNN, mean-only B.N. | 8.52% |
| our CNN, weight norm. | 8.46% |
| our CNN, normal param. | 8.43% |
| our CNN, batch norm. | 8.05% |
| ours, W.N. + mean-only B.N. | 7.31% |
| DenseNet [13] | 5.77% |

Figure 2: Classification results on CIFAR-10 without data augmentation.

epochs using the different parameterizations, we evaluate their performance on the CIFAR-10 test set. The results are summarized in table 2: weight normalization, the normal parameterization, and mean-only batch normalization have similar test accuracy ($\approx 8.5\%$ error). Batch normalization does significantly better at $8.05\%$ error. Mean-only batch normalization combined with weight normalization has the best performance at $7.31\%$ test error, and interestingly does much better than mean-only batch normalization combined with the normal parameterization: This suggests that the noise added by batch normalization can be useful for regularizing the network, but that the reparameterization provided by weight normalization or full batch normalization is also needed for optimal results. We hypothesize that the substantial improvement by mean-only B.N. with weight normalization over regular batch normalization is due to the distribution of the noise caused by the normalization method during training: for mean-only batch normalization the minibatch mean has a distribution that is approximately Gaussian, while the noise added by full batch normalization during training has much higher kurtosis. The result with mean-only batch normalization combined with weight normalization represented the state-of-the-art for CIFAR-10 among methods that do not use data augmentation, until it was recently surpassed by DenseNets [13].

## 5.2 Generative Modelling: Convolutional VAE

Next, we test the effect of weight normalization applied to deep convolutional variational auto-encoders (CVAEs) [16, 28, 29], trained on the MNIST data set of images of handwritten digits and the CIFAR-10 data set of small natural images.

Variational auto-encoders are generative models that explain the data vector $\mathbf{x}$ as arising from a set of latent variables $\mathbf{z}$, through a joint distribution of the form $p(\mathbf{z}, \mathbf{x}) = p(\mathbf{z})p(\mathbf{x}|\mathbf{z})$, where the *decoder* $p(\mathbf{x}|\mathbf{z})$ is specified using a neural network. A lower bound on the log marginal likelihood $\log p(\mathbf{x})$ can be obtained by approximately inferring the latent variables $\mathbf{z}$ from the observed data $\mathbf{x}$ using an *encoder* distribution $q(\mathbf{z}|\mathbf{x})$ that is also specified as a neural network. This lower bound is then optimized to fit the model to the data.

We follow a similar implementation of the CVAE as in [29] with some modifications, mainly that the encoder and decoder are parameterized with ResNet [11] blocks, and that the diagonal posterior is replaced with a more flexible specification based on *inverse autoregressive flow*. A further developed version of this model is presented in [17], where the architecture is explained in detail.

For MNIST, the encoder consists of 3 sequences of two ResNet blocks each, the first sequence acting on 16 feature maps, the others on 32 feature maps. The first two sequences are followed by a 2-times subsampling operation implemented using $2 \times 2$ stride, while the third sequence is followed by a fully connected layer with 450 units. The decoder has a similar architecture, but with reversed direction. For CIFAR-10, we used a neural architecture with ResNet units and multiple intermediate stochastic layers. We used Adamax [15] with $\alpha = 0.002$ for optimization, in combination with

Polyak averaging [26] in the form of an exponential moving average that averages parameters over approximately 10 epochs.

In figure 3, we plot the test-set lower bound as a function of number of training epochs, including error bars based on multiple different random seeds for initializing parameters. As can be seen, the parameterization with weight normalization has lower variance and converges to a better optimum. We observe similar results across different hyper-parameter settings.

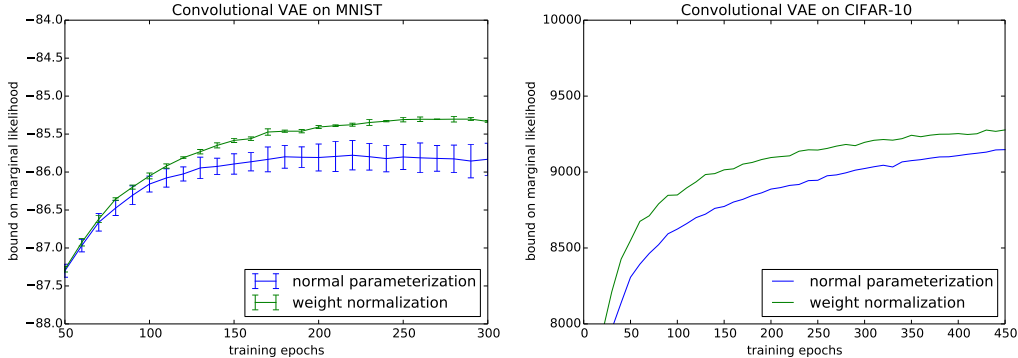

Figure 3: Marginal log likelihood lower bound on the MNIST (top) and CIFAR-10 (bottom) test sets for a convolutional VAE during training, for both the *standard* implementation as well as our modification with *weight normalization*. For MNIST, we provide standard error bars to indicate variance based on different initial random seeds.

## 5.3 Generative Modelling: DRAW

Next, we consider DRAW, a recurrent generative model by [9]. DRAW is a variational auto-encoder with generative model $p(\mathbf{z})p(\mathbf{x}|\mathbf{z})$ and encoder $q(\mathbf{z}|\mathbf{x})$, similar to the model in section 5.2, but with both the encoder and decoder consisting of a recurrent neural network comprised of Long Short-Term Memory (LSTM) [12] units. LSTM units consist of a memory cell with additive dynamics, combined with input, forget, and output gates that determine which information flows in and out of the memory. The additive dynamics enables learning of long-range dependencies in the data.

At each time step of the model, DRAW uses the same set of weight vectors to update the *cell states* of the LSTM units in its encoder and decoder. Because of the recurrent nature of this process it is not trivial to apply batch normalization here: Normalizing the cell states diminishes their ability to pass through information. Fortunately, weight normalization can easily be applied to the weight vectors of each LSTM unit, and we find this to work well empirically. Some other potential solutions were recently proposed in [4, 2].

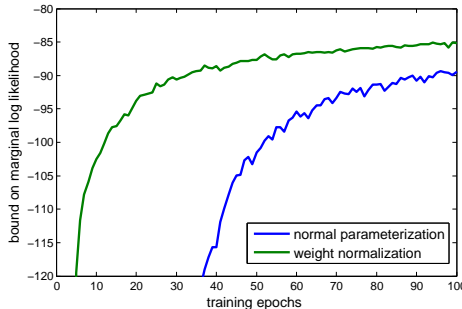

Figure 4: Marginal log likelihood lower bound on the MNIST test set for DRAW during training, for both the *standard* implementation as well as our modification with *weight normalization*. 100 epochs is not sufficient for convergence for this model, but the implementation using weight normalization clearly makes progress much more quickly than with the standard parameterization.

We take the Theano implementation of DRAW provided at `https://github.com/jbornschein/draw` and use it to model the MNIST data set of handwritten digits. We then make a single modification to the model: we apply weight normalization to all weight vectors. As can be seen in figure 4, this significantly speeds up convergence of the optimization procedure, even without modifying the initialization method and learning rate that were tuned for use with the normal parameterization.

## 5.4 Reinforcement Learning: DQN

Next we apply weight normalization to the problem of Reinforcement Learning for playing games on the Atari Learning Environment [3]. The approach we use is the Deep Q-Network (DQN) proposed by [25]. This is an application for which batch normalization is not well suited: the noise introduced by estimating the minibatch statistics destabilizes the learning process. We were not able to get batch normalization to work for DQN without using an impractically large minibatch size. In contrast, weight normalization is easy to apply in this context, as is the initialization method of section 3. Stochastic gradient learning is performed using Adamax [15] with momentum of 0.5. We search for optimal learning rates in $\{0.0001, 0.0003, 0.001, 0.003\}$, generally finding 0.0003 to work well with weight normalization and 0.0001 to work well for the normal parameterization. We also use a larger minibatch size (64) which we found to be more efficient on our hardware (Amazon Elastic Compute Cloud `g2.2xlarge` GPU instance). Apart from these changes we follow [25] as closely as possible in terms of parameter settings and evaluation methods. However, we use a Python/Theano/Lasagne reimplementation of their work, adapted from the implementation available at `https://github.com/spragunr/deep_q_rl`, so there may be small additional differences in implementation.

Figure 5 shows the training curves obtained using DQN with the standard parameterization and with weight normalization on Space Invaders. Using weight normalization the algorithm progresses more quickly and reaches a better final result. Table 6 shows the final evaluation scores obtained by DQN with weight normalization for four games: on average weight normalization improves the performance of DQN.

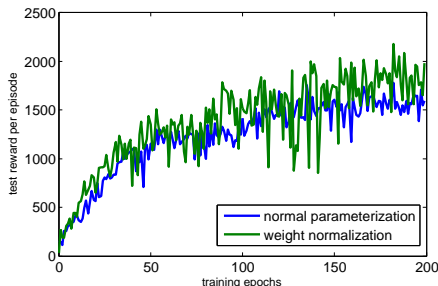

| Game | normal | weightnorm | Mnih |
|---|---|---|---|
| Breakout | 410 | 403 | 401 |
| Enduro | 1,250 | 1,448 | 302 |
| Seaquest | 7,188 | 7,375 | 5,286 |
| Space Invaders | 1,779 | 2,179 | 1,975 |

Figure 6: Maximum evaluation scores obtained by DQN, using either the normal parameterization or using weight normalization. The scores indicated by *Mnih et al.* are those reported by [25]: Our normal parameterization is approximately equivalent to their method. Differences in scores may be caused by small differences in our implementation. Specifically, the difference in our score on Enduro and that reported by [25] might be due to us not using a play-time limit during evaluation.

Figure 5: Evaluation scores for Space Invaders obtained by DQN after each epoch of training, for both the standard parameterization and using weight normalization. Learning rates for both cases were selected to maximize the highest achieved test score.

## 6 Conclusion

We have presented *weight normalization*, a simple reparameterization of the weight vectors in a neural network that accelerates the convergence of stochastic gradient descent optimization. Weight normalization was applied to four different models in supervised image recognition, generative modelling, and deep reinforcement learning, showing a consistent advantage across applications. The reparameterization method is easy to apply, has low computational overhead, and does not introduce dependencies between the examples in a minibatch, making it our default choice in the development of new deep learning architectures.

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
