[Supplementary Material]

# A  Neural network architecure for CIFAR-10 experiments

| Layer type | # channels | $x, y$ dimension |
|---|---|---|
| raw RGB input | 3 | 32 |
| ZCA whitening | 3 | 32 |
| Gaussian noise $\sigma = 0.15$ | 3 | 32 |
| $3 \times 3$ conv leaky ReLU | 96 | 32 |
| $3 \times 3$ conv leaky ReLU | 96 | 32 |
| $3 \times 3$ conv leaky ReLU | 96 | 32 |
| $2 \times 2$ max pool, str. 2 | 96 | 16 |
| dropout with $p = 0.5$ | 96 | 16 |
| $3 \times 3$ conv leaky ReLU | 192 | 16 |
| $3 \times 3$ conv leaky ReLU | 192 | 16 |
| $3 \times 3$ conv leaky ReLU | 192 | 16 |
| $2 \times 2$ max pool, str. 2 | 192 | 8 |
| dropout with $p = 0.5$ | 192 | 8 |
| $3 \times 3$ conv leaky ReLU | 192 | 6 |
| $1 \times 1$ conv leaky ReLU | 192 | 6 |
| $1 \times 1$ conv leaky ReLU | 192 | 6 |
| global average pool | 192 | 1 |
| softmax output | 10 | 1 |

Table 1: Neural network architecture for CIFAR-10.