[Reviews · NeurIPS 2016]

Reviewer 1

Summary

Inspired by batch-normalisation, the paper proposes to reparametrise neural networks by decoupling length and direction of the connection wight vectors. A theoretical analysis shows that the resulting gradients rescale the standard wight gradient and project it onto the compliment of the wight vector. It is agued that this leads to an improvement of the conditioning of the gradient. Experiments show that the proposed reparametrisation leads to competitive results on various bench mark sets and improves the performance of different NN models.

Qualitative Assessment

The suggested reparametrisation and its theoretical analysis are very interesting and I enjoyed reading the paper. However, some points in the theoretical analysis could be improved: The paper argues that the new parametrisation improves the conditioning matrix of the gradient, but neither a strong theoretical argument nor a empirical demonstration for this are given. In line 127 it is said "Empirically, we find that w is often (close to) a dominant eigenvector of the covariance matrix C", but the correspond experiments are neither shown in the paper nor in the supplemental material. In line 122/123 the authors claim "It has been observed that neural networks with batch normalization also have this property (to be relatively insensitive to different learning rates), which can be explained by this analysis.". However, it did not became clear to me, how the analysis of the previous sections can be directly transferred to batch normalisation. Regarding the experimental part: Except from the experiments on MNIST in section 5.2 each experiment is only run once (i.e. with one random initialisation). This makes an proper experimental evaluation of the different models challenging. Did the authors made sure that the observed differences were not only observes for this specific initialisation and was the same initialisation shared by all models? Several reported runs would moreover allow for statements about statistical significant performance differences. Why were all different methods (including the different versions of bath-normalisation) only compared in the first experiment, while the further compared only weight-normalisation against the standard parametrisation? It would be interesting to see how the other methods performed on this tasks as well. Often for the standard parametrisation a smaller learning rate is chosen than for weigh-normalisation? Couldn't this result in the faster learning? And did larger learning rates lead to to large variance for the standard parametrisation? Furthermore, the effective learning rates are probably different as well. It could be interesting to scale all gradients to the same length to analyse the pure effect of the changed direction of the gradients. Where the results reported in Figure 6 gained after 200 epochs? Would have further training improved the results? Missing references: When introducing mean-only batch normalisation the following paper could have been cited: Y. LeCun, L. Bottou, G. Orr, and K. R. Mu ̈ller. Efficient backprop. In Neural Networks: Tricks of the Trade, Lecture Notes in Computer Science, page 546. Springer Berlin/Heidelberg, 1998. In line 268/269 it says "it is not clear how batch normalization could be applied to this model (LSTM)". Note, that recent research by Cooijmans et all (2016) on "Recurrent Batch Normalisation" tries to answer this question (https://arxiv.org/pdf/1603.09025) minor comments: - in equation 9: w’ is not defined, in my opinion it took me a while to figure out that you mean the transposed vector (not the derivative), so that ww' is an outer product. -line 127: "gradient in v" -> gradient with respect to v -line 266: "enables" -> enable - Figure2 and 6 are Tables

Confidence in this Review

3-Expert (read the paper in detail, know the area, quite certain of my opinion)


Reviewer 2

Summary

This paper proposes a simple re-parameterization of neural network layers to speed up convergence. The idea is to represent a weight vector as the product of a scalar parameter times a vector divided by its norm/. Optimizing using such re-parameterization yields faster convergence across a variety of tasks. The authors propose also a variant which combines this re-parameterization with mean-only batch normalization, further improving results on CIFAR-10.

Qualitative Assessment

I overall enjoyed reading this paper. The idea is simple, well presented and it seems effective. Suggestions/questions: 1) without using the exponential form, how did the author make sure that "g" is positive? 2) how did the authors extend the reparameterization to convolutional layers? Is the normalization applied to all kernel parameters contributing to one output feature map, for instance? Please, clarify. 3) the experimental section could be strengthened with: - experiments on ImageNet - comparing to other approaches like [3] and [19] on at least two tasks - experiments proving the claimed improved robustness to learning rate choice - experiments using BN mean only + weight normalization on tasks other than CIFAR-10 classification - showing plots of training/test accuracy VS wall clock time -------------------------------- I have read the rebuttal and I think the authors have partially addressed my concerns. I hope they will keep their promise to update the paper and I have increased the score accordingly.

Confidence in this Review

2-Confident (read it all; understood it all reasonably well)


Reviewer 3

Summary

This paper introduces weight normalization, a reparameterization of neural network layers that provides an alternative to batch normalization. It has the same goal of speeding up convergence, but it achieves this with lower computational overhead and without introducing dependencies between examples in minibatches, which means it avoids the coupling between regularization and other hyperparameters (i.e. batch size) that batch normalization exhibits. This in turn makes it more suitable to use in noise-sensitive settings such as generative modelling and deep reinforcement learning. The method is evaluated in these settings (as well as for supervised learning) and shown to consistently improve convergence, and sometimes performance as well.

Qualitative Assessment

The paper provides a simple, easy to implement alternative to batch normalization which addresses some of the concerns that many people have about it, particularly its stochasticity (which is sometimes touted as an advantage because it has a regularizing effect, but can often be an annoyance as well). Weight normalization also provides easier inference: it does not require computing separate inference-time statistics, networks trained with weight normalization can simply be used as is. Two reference implementations are provided, which will further aid its adoption. Anecdotally, batch normalization is sometimes used in some parts of generative adversarial networks, but not in others (e.g. not in the discriminator). It would be very interesting to include such a setting as a third generative modelling experiment. That said, the experimental section of the paper is already very strong, and is definitely sufficient as it is. Are there any references for batch norm being unsuitable for RL? If so, it would be useful to include some at the end of Section 2.2. It is interesting that the authors find it useful to have the norm of v grow as learning progresses. The analysis in the second half of Section 2.1 and in Section 2.2 is very interesting, but this leaves me wondering: what would happen if the formulation was unchanged, but an additional normalisation step (v := v / ||v||) was inserted after every update? (Note that we would still backprop through v / ||v|| even though ||v|| is 1.) Would this negatively affect convergence? I'm also curious about the interaction with weight decay. There is no mention of weight decay in the paper as far as I recall, so I'm assuming it was not used in any of the experiments. But since this is a very common thing to do, it would be good to have some experiments that include it as well. If I'm not mistaken this could be implemented simply by decaying g and leaving v untouched, which is interesting in itself. Decaying v does not seem sensible considering the observation that having v grow during training helps convergence, but maybe the effect of this could also be investigated. In Section 5.2, a ResNet architecture is described with "multiple intermediate stochastic layers", but this is not really explained anywhere beyond a footnote that indicates a manuscript is in preparation. Some more details about what this looks like will be needed. Just to clarify: I'm guessing that mean-only BN was used only for the supervised experiments? All other experiments were done with WN only? It would be good to stress this to avoid any confusion. In general it would be good to explicitly show how crucial mean-only BN and the proposed initialisation scheme are for each result, so that WN itself can be judged in isolation.

Confidence in this Review

3-Expert (read the paper in detail, know the area, quite certain of my opinion)